# Beneficial Effects of Ginger Root Extract on Pain Behaviors, Inflammation, and Mitochondrial Function in the Colon and Different Brain Regions of Male and Female Neuropathic Rats: A Gut–Brain Axis Study

**DOI:** 10.3390/nu16203563

**Published:** 2024-10-21

**Authors:** Julianna Maria Santos, Hemalata Deshmukh, Moamen M. Elmassry, Vadim Yakhnitsa, Guangchen Ji, Takaki Kiritoshi, Peyton Presto, Nico Antenucci, Xiaobo Liu, Volker Neugebauer, Chwan-Li Shen

**Affiliations:** 1Department of Pathology, Texas Tech University Health Sciences Center, Lubbock, TX 79430, USA; julianna.santos@ttuhsc.edu (J.M.S.); phsdeshmukh@gmail.com (H.D.); xiaobo.liu@ttuhsc.edu (X.L.); 2Department of Microanatomy and Cellular Biology, Woody L. Hunt School of Dental Medicine, Texas Tech University Health Sciences Center, El Paso, TX 79905, USA; 3Department of Molecular Biology, Princeton University, Princeton, NJ 08540, USA; elmassry@princeton.edu; 4Department of Pharmacology and Neuroscience, Texas Tech University Health Sciences Center, Lubbock, TX 79430, USA; vadim.yakhnitsa@ttuhsc.edu (V.Y.); guangchen.ji@ttuhsc.edu (G.J.); takaki.kiritoshi@ttuhsc.edu (T.K.); peyton.presto@ttuhsc.edu (P.P.); nico.antenucci@ttuhsc.edu (N.A.); volker.neugebauer@ttuhsc.edu (V.N.); 5Center of Excellence for Translational Neuroscience and Therapeutics, Texas Tech University Health Sciences Center, Lubbock, TX 79430, USA; 6Garrison Institute on Aging, Texas Tech University Health Sciences Center, Lubbock, TX 79430, USA; 7Center of Excellence for Integrative Health, Texas Tech University Health Sciences Center, Lubbock, TX 79430, USA; 8Obesity Research Institute, Texas Tech University, Lubbock, TX 79401, USA

**Keywords:** bioactive compounds, pain, central nervous system, leaky gut, mitochondrial function, neuroimmune signaling, animals

## Abstract

Background: Neuroinflammation and mitochondrial dysfunction have been implicated in the progression of neuropathic pain (NP) but can be mitigated by supplementation with gingerol-enriched ginger (GEG). However, the exact benefits of GEG for each sex in treating neuroinflammation and mitochondrial homeostasis in different brain regions and the colon remain to be determined. Objective: Evaluate the effects of GEG on emotional/affective pain and spontaneous pain behaviors, neuroinflammation, as well as mitochondria homeostasis in the amygdala, frontal cortex, hippocampus, and colon of male and female rats in the spinal nerve ligation (SNL) NP model. Methods: One hundred rats (fifty males and fifty females) were randomly assigned to five groups: sham + vehicle, SNL + vehicle, and SNL with three different GEG doses (200, 400, and 600 mg/kg BW) for 5 weeks. A rat grimace scale and vocalizations were used to assess spontaneous and emotional/affective pain behaviors, respectively. mRNA gene and protein expression levels for tight junction protein, neuroinflammation, mitochondria homeostasis, and oxidative stress were measured in the amygdala, frontal cortex, hippocampus, and colon using qRT-PCR and Western blot (colon). Results: GEG supplementation mitigated spontaneous pain in both male and female rats with NP while decreasing emotional/affective responses only in male NP rats. GEG supplementation increased intestinal integrity (claudin 3) and suppressed neuroinflammation [glial activation (GFAP, CD11b, IBA1) and inflammation (TNFα, NFκB, IL1β)] in the selected brain regions and colon of male and female NP rats. GEG supplementation improved mitochondrial homeostasis [increased biogenesis (TFAM, PGC1α), increased fission (FIS, DRP1), decreased fusion (MFN2, MFN1) and mitophagy (PINK1), and increased Complex III] in the selected brain regions and colon in both sexes. Some GEG dose–response effects in gene expression were observed in NP rats of both sexes. Conclusions: GEG supplementation decreased emotional/affective pain behaviors of males and females via improving gut integrity, suppressing neuroinflammation, and improving mitochondrial homeostasis in the amygdala, frontal cortex, hippocampus, and colon in both male and female SNL rats in an NP model, implicating the gut–brain axis in NP. Sex differences observed in the vocalizations assay may suggest different mechanisms of evoked NP responses in females.

## 1. Introduction

Neuropathic pain (NP) is pain originating from a lesion or disease affecting the somatosensory nervous system in 7–10% of the world population [1]. The effects of nerve injury include neuroinflammation and neuroplastic changes in peripheral neurons and central neurons associated with sensitization and hyperexcitability [2]. NP leads to mechanical/thermal hypersensitivity stemming from nociceptive processes (i) in the peripheral nervous system (PNS) with the production of inflammatory mediators from injured or adjacent tissue and nerve fibers and (ii) in the central nervous system (CNS) involving excessive oxidative stress production, microglial activation [3], and mitochondrial dysfunction, leading to proinflammatory cytokine/chemokine signaling [2,3,4,5,6,7].

In neuroinflammation, glial activation and mitochondrial dysfunction in the PNS and CNS cause the production of proinflammatory cytokines/chemokines. Certain regions of the brain, namely the medial prefrontal cortex (mPFC), amygdala, and hippocampus, play crucial roles in modulating neuroinflammation during chronic NP development [8,9,10]. For example, peripheral nerve injury-induced NP suppresses pyramidal cell firing in the prelimbic area of the mPFC as a result of feed-forward inhibition, mediated by parvalbumin-expression GABAergic interneurons [8]. Recent research has also shown the amygdala’s central role in the emotional/affective spectrum of pain/pain modulation. Hyperactivity in the laterocapsular division of the central nucleus of the amygdala, also called the “nociceptive amygdala”, is responsible for pain-associated emotional reactions and anxiety-like behavior [9]. The hippocampus is activated during pain processing and modulation of nociceptive stimuli [10]. In light of these observations, targeting of glial activation, mitochondrial dysfunction, and proinflammatory cytokine and chemokine signaling in the mPFC, amygdala, and hippocampus in order to suppress neuroinflammation may bear promise in alleviating or preventing NP states.

Recent discoveries point to a connection between gut microbiota and CNS disorders and between NP and gut microbiota modifications. Immune mediators, gut-derived metabolites, and nervous structures enable crosstalk between gut microbiota, the PNS, and the CNS [11]. Measurements of inflammatory mediators, metabolic products, or neurotransmitters show how the gut–CNS connection operates [12]. In a pre-clinical spinal nerve ligation (SNL)-induced NP study, an SNL procedure resulted in an increased urinary lactulose/D-mannitol ratio (a marker of intestinal permeability) and increased NFκB and TNFα gene expression in the colon and amygdala of male rats with NP, suggesting the link between the gut and brain [13]. However, no study has investigated the connection between the colon and other regions of the brain, such as the mPFC and hippocampus.

Ginger (*Zingiber officinale* Rosc., belongs to Zingiberaceae family), also known as adrak (Hindi), Jengibre (Spanish), Jiang (Chinese), or Zanjabeel (Arabic) is a perennial herbaceous plant that is grown across the Indian subcontinent and Japan and Indonesia, as well as in countries like Brazil and Jamaica [14]. Ginger is best known for its culinary and medicinal uses [15]. The rhizome of ginger is marketed in various forms, such as dry ginger, dry ginger powder, raw ginger, ginger oil, and ginger oleoresin [16]. These products find their use in a variety of foods, including ginger candy, ginger beer, ginger squash, ginger flakes, pickles, and sweet vinegar, or just as raw ginger powder. Ginger is also well known in the traditional medicinal systems of China (traditional Chinese medicine, TCM), India (Ayurveda), and the Middle East and Africa. India is the world’s largest exporter of ginger powder; ginger is cultivated across the country, with the northeastern belt of India representing a major ginger growing area [14].

Ginger and its bioactive components can penetrate the blood–brain barrier (BBB) via passive diffusion, suggesting that ginger could have positive effects on the CNS [17]. Ginger can attenuate the gene/protein expression levels of inducible nitric oxide synthase (iNOS), cyclooxygenase-2 (COX-2), and proinflammatory cytokines in lipopolysaccharide (LPS)-stimulated microglial cells through the mitogen-activated protein kinase (MAPK) and nuclear factor kappa B (NFκB) signaling cascades [18]. 6-Shogaol (the most abundant bioactive compound in ginger) suppresses LPS-induced microglial activation in both neuroglia culture and animals with neuroinflammation. 6-Shogaol significantly inhibits iNOS expression and nitric oxide production in microglial cell culture, via inhibiting gene expression of COX-2, NFκB, and MAPK as well as the production of prostaglandin E_2_, tumor necrosis factor-alpha (TNFα), and interleukin 1beta (IL-1β). 6-Shogaol (also a bioactive compound in ginger) exhibits neuroprotective effects in transient global ischemia by hindering microglial activity [19]. Moreover, ginger can improve mitochondrial function by stopping mitochondrial apoptosis, enhancing the expression of antioxidant genes (i.e., glutathione peroxidase, superoxide dismutase-1, and catalase), and decreasing oxidant factors (lipid peroxidation, ROS, caspase-3 activity, and the Bax/Bcl-2 ratio) [20,21]. We recently reported that gingerol-enriched ginger (GEG) improves glucose homeostasis in rats with diabetes by improving GI mitochondrial function and intestinal integrity [22]. We also demonstrated that GEG reduced mechanical hypersensitivity, pain-associated anxio-depressive behavior, and proinflammation of neuroimmune cells in diabetic rats with neuropathy [23]. GEG supplementation significantly alleviated pain-associated behaviors (mechanosensitivity, emotional pain, and spontaneous pain) and reduced SNL-induced intestinal permeability and neuroinflammation in male SNL rats [24]. We recently reported that ginger polyphenols can reverse amygdala neuroimmune signaling and modulate microbiome composition in the molecular signature of NP rats [25]. GEG-treated NP animals had a reduced abundance of *Muribaculaceae*, *Clostridia UCG-014*, *Rikenella*, *RF39*, *Mucispirillum schaedleri*, *Clostridia UCG-009*, and *Acetatifactor*, while they had an increased abundance of *Anaerofustis stercorihominis*, *Clostridium innocuum* group, *Hungatella*, and *Flavonifactor* [25].

Current evidence indicates a strong link between sex and pain modulation [26]. Data in the literature show marked differences in men’s and women’s responses to pain, with women demonstrating more variability and higher pain sensitivity, and more often reporting pain-associated diseases than men [27]. Factors affecting pain hypersensitivity in women and men include differing sex hormones and modulation of the endogenous opioid system [27]. Female animals may show differences in pain behaviors, pain-associated affective behaviors, biochemical profiles, and responsiveness to therapies/treatments. For example, in rats with partial sciatic nerve ligation (PSNL)-induced NP, the female rats were significantly more likely to evolve NP than male rats [28]. In rats with chronic SNL-induced NP, female rats showed greater increases in vocalizations (emotional pain indicator), anxiety-like behaviors, and emotional/affective responses than male rats [29]. We previously reported that the beneficial effects of GEG on brain and gut function and NP-related behaviors make the gut–brain axis an important target of GEG, as we have previously shown with GEG reversing the molecular signature of amygdala neuroimmune signaling and modulating the microbiome of male rats with NP [25].

The effects of GEG on chronic NP in connection with molecular mechanisms have never been studied previously with regard to differences between the sexes. Therefore, this study examined the effects of three different GEG doses on NP-related behaviors and intestinal integrity in both male and female rats. We also investigated the impacts of GEG on neuroinflammation and mitochondrial homeostasis in the animals. We selected three regions of the brain (amygdala, mPFC, and hippocampus) and colon to explore the connection between the CNS and gut in chronic NP. We hypothesized that GEG administration to the NP animals would mitigate NP-related sensory and affective behaviors, alleviate neuroinflammation signaling, and improve mitochondrial function and intestinal integrity. Such results would speak strongly for the existence of the CNS–gut axis.

## 2. Materials and Methods

### 2.1. Animals and Treatments

One hundred Sprague Dawley rats (50 males and 50 females, 140–170 g, Envigo, Cumberland, VA, USA) were housed individually and acclimated for 5 days to a 12 h light/dark cycle. Food (AIN-93G diet, Research Diet, Inc., New Brunswick, NJ, USA) and water were provided ad libitum. This study, along with all procedures, was approved by the Texas Tech University Health Sciences Center Institutional Animal Care and Use Committee (IACUC protocol number: 20032, approved on 31 January 2021).

After acclimatization, 10 male and 10 female rats underwent a sham procedure while the remaining 40 male and 40 female rats received the SNL procedure, which is commonly used for studying NP mechanisms prior to clinical trials [23,24,25,26,30,31,32,33]. SNL at the left L4–L5 position in the rats leads to acute pain hypersensitivity within 7 days and lasts for a couple of months [34]. In the sham group, the left L4 and L5 spinal nerve was exposed without ligation.

We randomly assigned rats into five treatment groups for both males and females, respectively: sham animals receiving corn oil, as the vehicle (Sham-V group), SNL animals receiving corn oil (SNL-V group), SNL animals receiving GEG (a gift from Sabinsa Corporation, East Windsor, NJ, USA) at 200 mg/kg BW (SNL + 200GEG group), 400 mg/kg BW (SNL + 400GEG group), and 600 mg/kg BW (SNL + 600GEG group). We administered the corn oil and GEG daily orally for 5 weeks. According to the results of gas chromatography–mass spectrometry, GEG consists of 18.7% 6-gingerol, 1.81% 8-gingerol, 2.86% 10-gingerol, 3.09% 6-shogoal, 0.39% 8-shogaol, and 0.41% 10-shogaol. Measurements of food intake, water consumption, and body weight were taken every week.

### 2.2. Pain Assessment

#### 2.2.1. Spontaneous Pain

We performed rat grimace scale (RGS) scoring at baseline (before surgery) and end of study (after 5-week intervention) based on our previous studies [35,36]. In brief, the rats were placed in individual plexiglass chambers with 2 video cameras to record for 10 min with still images being chosen (10 representative images with a number code) using customized Python 3.13.0 scripts. Based on 4 parameters, orbital tightening, ear position, nose bulge, and whisker change, 3 blinded experienced evaluators performed RGS scoring from 0 to 2 (0 = not present, 1 = moderate, 2 = severe). We calculated a total score as the sum of the 4 parameters for each evaluator and the average across the 3 evaluators for 4 parameters to obtain the final RGS score.

#### 2.2.2. Emotional Pain Responses

To access emotional pain responses, we measured vocalizations at one day before the SNL/sham procedure (baseline) and after intervention, according to our previous studies [29,37,38,39]. In brief, after 10 min for rats to adjust the recording chamber, vocalizations were evoked with a calibrated forceps by innocuous and noxious stimuli to the left hind paw and recorded for 1 min at a time beginning when mechanical stimulation was applied. The resulting signals were digitized and analyzed with an UltraVox interface (Noldus Information Technology, Leesburg, VA, USA). We measured vocalizations twice within ten minutes for each animal and averaged the scores.

### 2.3. Collection of Samples

After 5-week feeding period, the animals were anesthetized and euthanized for blood, brain, and colon collection. Plasma was obtained and stored at −80 °C for later lipopolysaccharide-binding protein (LBP) measurement. Brain regions under study [frontal cortex (right, medial prefrontal cortex), amygdala (right central nucleus), and hippocampus (right, dorsal)] and the colon (distal) were harvested, immediately snap-frozen in liquid nitrogen, and kept at −80 °C for later analysis of gene and protein expression (colon only).

### 2.4. Intestinal Permeability Measurement

Intestinal permeability was determined by measuring the concentrations of plasma LBP with an ELISA kit (catalog number: LS-F21745-1, Life Span Biosciences, Inc., Shirley, MA, USA) as per the manufacturer’s instruction.

### 2.5. RNA Isolation and qRT-PCR

Total RNA was isolated from the amygdala (right), frontal cortex (right), hippocampus (right), and colon (distal) based on our published work [22,23]. Each transcribed cDNA was used to amplify the targeted gene using the respective primers (Table 1). The gene expression levels of β-actin were used to normalize respective gene expression and calculated using the following formula: 2^−(ΔCT×1000)^ [40].

### 2.6. Western Blot Analysis

To validate the PCR data at the protein level, we selected the colon for Western blot analysis as it offered enough tissue to perform Western blot analysis. Protein expression levels were determined with their respective primary and secondary antibodies (Table 2) according to our previous work [41]. The protein expression levels of β-actin were used to normalize the respective protein expression. We performed densitometric analyses using ImageJ-win64 software.

### 2.7. Statistical Analysis

Sample size calculation: Previous and preliminary data [30,31] indicate that to obtain significance at the α = 0.05 at statistical power 0.9, *n* = 6–10 animals pre group are necessary to detect difference in vocalizations using GraphPad Prism 9 (GraphPad Software, San Diego, CA, USA). Results of findings are listed as the mean ± standard error of the mean (SEM). For the various experimental groups, we included total RGS score, total duration of vocalization for one minute, and plasma LBP concentrations by one-way or two-way ANOVA followed by post hoc tests (GraphPad Software, San Diego, CA, USA). We measured gene and protein expression levels by one-way ANOVA followed by post hoc tests at *p* < 0.05.

## 3. Results

### 3.1. GEG Reduced Spontaneous Pain in Both Sexes

At baseline, we detected no significant differences in the total RGS score across all groups of rats with NP. At the end of the study, both male and female rats that had received GEG supplementation had lower levels of SNL-induced spontaneous pain (Figure 1A for male rats and Figure 1B for female rats), as shown in the total RGS scores, regardless of GEG doses.

### 3.2. GEG Reduced Emotional Pain Only in Male NP Rats

At baseline, we detected no significant differences in the length of audible and ultrasonic vocalizations across all male and female rats with NP (*p* > 0.05). At the study’s conclusion, the SNL-V group had significantly increased ultrasonic vocalizations in male (but not female) NP rats in response to innocuous and noxious mechanical stimuli compared to those in the Sham-V group (Figure 2A for male rats and Figure 2B for female rats). GEG supplementation significantly mitigated SNL-induced emotional pain responses in male (but not female) NP rats, as reflected in decreased audible and ultrasonic vocalizations evoked by noxious mechanical stimuli, regardless of GEG doses (Figure 2A,B).

### 3.3. GEG Improved Intestinal Permeability and Tight Junction Protein Integrity

Regarding intestinal permeability, there was an increase in plasma LBP levels in female (but not male) NP rats five weeks after SNL induction relative to the sham rats (Figure 3A for male rats and Figure 3B for female rats). GEG supplementation significantly decreased plasma LBP levels in male NP rats (only SNL + 200GEG group) and female NP rats (all three GEG-supplemented groups), suggesting decreased intestinal permeability (Figure 3A,B).

Figure 4 shows the gene expression levels of claudin 3 (a tight junction protein marker) in the amygdala, frontal cortex, hippocampus, and colon in male (Figure 4A) and female (Figure 4B) NP rats. Compared to the sham operation, the SNL operation yielded lower claudin 3 gene expression levels in the colon of male rats and in the amygdala of female rats. In male NP rats, GEG administration significantly increased claudin 3 gene and protein expression levels in the colon (SNL + 200GEG and SNL + 400GEG) and claudin 3 gene expression levels in the frontal cortex (SNL + 400GEG and SNL + 600GEG) and hippocampus (SNL + 200GEG). In female NP rats, GEG administration increased claudin 3 gene expression levels in the amygdala, but not in the frontal cortex, hippocampus, and colon. While GEG supplementation did lead to an increase in claudin 3 mRNA expression levels in the colon of female NP rats, this effect was not statistically significant. Next, we used Western blot analysis of claudin 3 protein expression in the colon to validate the qRT-PCR data for males and resolve the non-significant trend in mRNA data for females. Figure 4C shows that GEG supplementation significantly increased claudin 3 protein expression in all rats with NP.

### 3.4. GEG Mitigated Neuroinflammation

The gene expression levels related to astrocyte activation, glial activation, immune response, and inflammation in the amygdala, frontal cortex, hippocampus, and colon are presented in Figure 5 for male rats and in Figure 6 for female rats.

We reported the gene expression levels of GFAP (astrocyte activation markers), CD11b, and IBA1 (microglial activation marker) in the designated brain regions and colon. In rats with NP (SNL), GFAP gene expression levels increased in the amygdala (males and females), frontal cortex (females), hippocampus (males and females), and colon (males), but decreased in the frontal cortex for males. GEG administration reversed SNL-associated changes in GFAP gene expression levels in the amygdala (all three GEG dosages in males; 200 and 600 mg/kg in females), frontal cortex (400 and 600 mg/kg in males; 200 and 600 mg/kg in females), hippocampus (200 mg/kg in males; 400 and 600 mg/kg in females), and colon (all three GEG dosages in males). Neither the SNL procedure nor GEG supplementation changed GFAP gene expression in the colon of female rats. CD11b gene expression levels were increased in the amygdala (males), frontal cortex (females), and colon (males and females) in SNL rats. GEG mitigated the SNL-induced changes in CD11b gene expression levels in the amygdala (males), frontal cortex (female), and colon (all three GEG dosages in males; 400 and 600 mg/kg in females). On the other hand, CD11b expression decreased in the amygdala (females) and hippocampus (males and females) of SNL rats. GEG increased CD11b gene expression levels in the amygdala (all three GEG dosages in females) and hippocampus (400 mg/kg in males; 400 mg/kg in females). In SNL rats, IBA1 gene expression was higher in the frontal cortex of females and in the colon of males. GEG decreased the IBA1 gene expression in the amygdala (all three GEG dosages in males and females) and colon (all three GEG dosages in males; 400 mg/kg in females). GEG increased the IBA1 gene expression in the frontal cortex of male NP rats (400 and 600 mg/kg). GEG did not significantly affect the IBA1 gene expression in the frontal cortex (females) and hippocampus (males and females) of NP rats.

We measured the gene expression levels of TLR4 and TLR2 (immune response) in the designated brain regions and colon. TLR4 gene expression levels in SNL rats decreased in the amygdala (males and females) but increased in the hippocampus (males). GEG administration reversed SNL-associated changes in TLR4 gene expression levels in the amygdala (200 mg/kg in males; 400 and 600 mg/kg in females), frontal cortex (all three GEG dosages in females), hippocampus (all three GEG dosages in males), and colon (400 and 600 mg/kg in males; 400 mg/kg in females). Regarding TLR 2, gene expression levels in SNL NP rats decreased in the amygdala (males and females), frontal cortex (males and females), and hippocampus (females), but increased in the colon (males and females). GEG administration reverted SNL-associated changes in TLR2 gene expression levels in the amygdala (200 and 400 mg/kg in males; 400 and 600 mg/kg in females), hippocampus (400 and 600 mg/kg in males), and colon (all three GEG dosages in males; 600 mg/kg in females). We assessed the gene expression levels of TNFα, NFκB, and IL1b (proinflammatory markers) in the collected tissue samples. Gene expression levels of TNFα, NFκB, and IL1b in SNL rats increased in the amygdala (males and females), frontal cortex (females), and colon (males and females). We found that the SNL operation did not significantly increase the gene expression of TNFα and NFκB in the frontal cortex and hippocampus of male rats, but it did induce significantly higher gene expression levels of TNFα and NFκB in the frontal cortex and that of NFκB in the hippocampus of female rats. As expected, GEG administration mitigated SNL-induced changes in TNFα, NFκB, and IL1b in the respective tissues of males and females with NP.

### 3.5. GEG Improved Mitochondrial Function

The gene expression levels for mitochondrial function (biogenesis, fission, fusion, respiratory chain complex, and mitophagy) in the amygdala, frontal cortex, hippocampus, and colon are presented in Figure 7 for male rats and Figure 8 for female rats.

We assessed the gene expression levels of TFAM and PGC1α in our tissue samples for mitochondrial biogenesis. Compared to the Sham-V group, male rats in the SNL-V group had increased TFAM gene expression in the amygdala and colon. Likewise, relative to the Sham-V group, female rats in the SNL-V group had increased TFAM gene expression in the hippocampus and colon, but decreased TFAM gene expression in the amygdala and frontal cortex. GEG supplementation increased TFAM gene expression in the amygdala (females in SNL + 600GEG), frontal cortex (males in SNL + 400GEG and SNL + 600GEG; females in all three GEG dosages), hippocampus (males in SNL + 400GEG and SNL + 600GEG), and suppressed TFAM gene expression in the hippocampus (females in SNL + 200GEG) and colon (males in all three GEG dosages and females in all three dosages). Regarding PGC1α, relative to the sham procedure, male rats that underwent the SNL procedure exhibited decreased PGC1α gene expression levels in the amygdala and colon, but increased levels of PGC1α in the hippocampus. Supplementation of GEG into the NP rats resulted in increased PGC1α gene expression in the amygdala (males in SNL + 600GEG) and colon (males in all three GEG dosages), but it suppressed that of gene expression in the hippocampus (males in SNL + 400GEG and SNL + 600GEG). Neither SNL procedure nor GEG supplementation affected PGC1α gene expression in the female rats, except for GEG at 400 mg/kg BW, which increased PGC1α expression in the amygdala.

Regarding mitochondrial fission markers, the SNL procedure decreased FIS1 gene expression in the colon of males, but it increased FIS1 gene expression in the hippocampus and colon of females. Supplementation of GEG significantly affected FIS1 gene expression by (i) increasing gene expression levels in the frontal cortex (males in SNL + 400GEG and SNL + 600GEG; females in all three GEG groups) and colon (males in SNL + 200GEG) and (ii) decreasing levels in the amygdala (females in all three GEG groups), hippocampus (males in SNL + 200GEG; females in 200GEG and 400GEG), and colon (females in SNL + 400GEG and SNL + 600GEG). Rats that underwent the SNL procedure exhibited suppressed DRP1 gene expression in the amygdala (males only), frontal cortex (males and females), and hippocampus (females), while showing increased DRP1 gene expression in the colon (males). In female NP rats, GEG’s inhibitory effects on DRP1 gene expression were observed in the amygdala (SNL + 400GEG) and GEG’s stimulating effects were found in the frontal cortex (SNL + 200GEG and SNL + 400GEG). Unlike with female NP rats, GEG stimulated DRP1 gene expression in male NP rats in the amygdala (SNL + 200GEG and SNL + 400GEG), frontal cortex and hippocampus (SNL + 400GEG and SNL + 600GEG), but suppressed gene expression in the colon (SNL + 200GEG).

In terms of mitochondrial fusion markers, SNL procedures had no effect on MFN2 gene expression in collected tissues from both males and females, except for decreased MFN2 gene expression in the frontal cortex of female rats. GEG supplementation resulted in elevated MFN2 gene expression in the amygdala (females in SNL + 200GEG and SNL + 400GEG), frontal cortex (males in SNL + 400GEG; females in SNL + 200GEG), and colon (females in SNL + 400GEG and SNL + 600GEG), but decreased MFN2 gene expression in the hippocampus (males in all GEG groups). Similar to MFN2 gene expression, among all tissue types from males and females, SNL procedures increased MFN1 gene expression only in the hippocampus (males), but decreased MFN1 gene expression in the frontal cortex and hippocampus (females). GEG administration significantly elevated MFN1 gene expression in the amygdala (males in SNL + 200GEG; females in SNL + 200GEG and SNL + 400GEG), frontal cortex (females in all three GEG groups), and colon (females in SNL + 400GEG), whereas it suppressed MFN1 gene expression in the hippocampus (males in SNL + 400GEG and SNL + 600GEG).

With regard to markers of the mitochondrial respiratory chain complex, relative to the Sham-V group, the SNL-V group exhibited lowered Complex III gene expression levels (i) in the hippocampus and colon of male rats, and (ii) in the amygdala and frontal cortex of female rats. After GEG administration to the NP rats, we observed increased Complex III gene expression levels (i) in the amygdala (all three GEG groups), frontal cortex and hippocampus (SNL + 400GEG and SNL + 600GEG), and colon (SNL + 400GEG) in male NP rats and (ii) in the amygdala (SNL + 600GEG) and frontal cortex (all three GEG groups) in female NP rats. Similar to Complex III, Complex I gene expression was suppressed by the SNL procedure in the amygdala (males and females). GEG supplementation increased Complex I gene expression in the collected tissues of both male and female NP rats.

PINK1 gene expression was assessed as a marker for mitophagy. SNL procedures induced higher PINK1 gene expressions in the amygdala (female only), frontal cortex (males and females), hippocampus (females only), and colon (males only). GEG supplementation suppressed SNL-induced PINK1 gene expression levels in the frontal cortex (males: SNL + 400GEG and SNL + 600GEG; females: all three GEG doses) and hippocampus (males: SNL + 400GEG and SNL + 600GEG; females: all three GEG doses). Neither SNL procedure nor GEG supplementation affected PINK1 gene expression in the colon of female rats.

## 4. Discussion

This study evaluated the impacts of GEG supplementation on spontaneous and emotional/affective pain behaviors as well as gut–brain health in male and female rats in a chronic NP model (SNL model). We previously reported that GEG supplementation significantly reduced spontaneous pain and emotional/affective pain behaviors in male rats with chronic NP; here, we have found that both male and female rats developed similar mechanical hypersensitivity after an SNL procedure, which agrees with another study showing no sex-based differences in chemotherapy-induced cold and mechanical allodynia [42], whereas female rats developed stronger mechanical allodynia than male rats after PSNL [43]. We also found that GEG had similar effects on hypersensitivity in male and female NP rats. On the other hand, we found increased emotional responses (stimulus-evoked vocalizations) in male, but not female, NP rats, and that GEG mitigated these effects only in males, indicating sex differences in emotional/affective pain. Our findings are consistent with patterns of male rats vocalizing more than female rats in fear conditioning [44]. Our previous study found vocalizations to be the most effective indicator of inter-individual and sex differences, though female NP rats exhibited significantly increased audible and ultrasonic vocalizations compared to male NP rats [29]. A larger sample size was used in the previous study, which may explain the significant differences between males and females in the vocalization assay. In our previous study [29], a larger sample size was used for the female study.

Emerging evidence suggests a connection between leaky gut and the development of chronic NP, which includes increased permeability of the intestinal mucosa and low-grade inflammation in the colon [45] and CNS [11,13,46,47,48]. A leaky gut can be caused by gut dysbiosis, resulting in inflammation, immune activation, and mood disorder [49]. In this study, oral GEG administration reduced spontaneous pain responses and intestinal permeability (as shown by increased plasma LBP) in male and female NP rats, suggesting that GEG may mitigate pain at least in part by improving intestinal integrity. Such findings corroborate and expand upon our previous study, which showed how GEG supplementation significantly mitigated spontaneous and emotional/affective pain and increased intestinal permeability in male NP rats, as shown by an increased ratio of lactulose to D-mannitol [50]. At the molecular level, increased protein expression of claudin 3 (a tight junction protein marker) in the colon of GEG-supplemented male and female NP rats indicates that GEG mitigates leaky gut. Downregulation of claudin 3 in the brain has been linked to increased permeability changes of the BBB and abnormal neuronal activity [51].

SNL-induced NP is associated with neuroinflammation [52,53,54], which can disrupt the BBB [55,56]. Males have a larger amygdala, a higher pain threshold, and lower pain sensitivity than females in humans. The authors also found that males exhibit less pain-related fear, which is correlated to differential amygdala function between males and females [57]. Moreover, BBB disruption has been observed in multiple mental disorders and was also shown to be related to sex differences [58]. Erickson et al. reported how sex-related factors impact BBB permeability due to LPS-induced neuroinflammation [59]. Consequently, the differential expression of claudin 3 in the male and female amygdala of the present study may suggest a connection between sex differences and pain-related negative emotions associated with SNL-induced neuroinflammatory BBB disruption. Intriguingly, in this study, GEG was able to attenuate the permeability of the BBB caused by SNL via increasing the expression of claudin 3 in female NP rats (Figure 4B). The effects of GEG on the claudin 3 gene expression appear to be sex- and tissue-specific, because they were more prominent in the amygdala of the females than that of the males. Interestingly, GEG had dose-dependent bidirectional effects on claudin 3 gene expression in the hippocampus of male NP rats, with facilitation at the low dose (200 mg/kg BW) and inhibition at the higher dosages (400 and 600 mg/kg BW).

Neuroinflammation involving microglia and astrocytes plays a major role in pain, including NP [4,60,61,62,63], which contributes to synaptic modulation and neuroplasticity. Current knowledge about neuroimmune signaling in pain status largely results from research on peripheral and spinal nociceptive processing. By contrast, little is known about similar processes in regions of the brain associated with various aspects of pain, including NP [2,64,65]. Neuroplasticity in the corticolimbic system, especially the amygdala, has become understood as a significant factor in the emotional/affective dimensions of pain and pain modulation [29]. To our knowledge, this study is the first to demonstrate that GEG supplementation can suppress SNL-induced GFAP and proinflammatory TNFα and IL-1β gene expression in the amygdala and hippocampus of both male and female NP rats. Previous studies showed that ginger extract alleviated morphine-induced neuroinflammation and GFAP activation in the nucleus accumbens of rats [66]. *Zingiber cassumunar* (*Z*. *cassumunar*) extract markedly reduced LPS-induced neuronal cell loss by suppressing GFAP activation and neuroinflammation in the hippocampus [67]. 6-Gingerol (the most abundant bioactive compound in ginger extract) attenuated increased levels of GFAP induced by LPS and TNFα in the rat brain [68]. The present study found sex-based differences in the effects of GEG on GFAP gene expression in the frontal cortex, where GEG had a stimulatory effect in male rats but was inhibitory in female NP rats. Interestingly, SNL induced bidirectional effects that differed between the sexes, where SNL suppressed GFAP expression in males and activated GFAP expression in females. In both males and females, GEG was able to redress this change and brought the GFAP expression back to normal levels. The neuroprotective effect of ginger on select regions of the brain (frontal cortex, dentate gyrus, and cerebellum) in male rats with diabetes has been linked to reducing oxidative stress, apoptosis, and inflammation as well as mitigating the astroglia response [69]. In brief, GEG exhibits differential effects on various regions of the brain based on sex. Astrocyte function is complex and can be protective in chronic pain, as we showed recently for the amygdala [70].

The differences in baseline GFAP expression and the effects of SNL on GFAP expression in males and females may be related to the different developmental timelines of the frontal cortex in males and females. The frontal lobe is known to be the last region of the brain to fully develop across various species, including rodents [71]. Rats mature between weeks 5 and 10, with female rats typically reaching puberty 10–20 days before males [72,73]. The earlier maturation of females is due to the differential rate of frontal cortex development [74]. The prefrontal cortex is critical in pain processing [8], and research has suggested that alterations in astrocyte activity, which was indicated as the GFAP expression level in this study (Figure 5A and Figure 6A), are involved in pain mechanisms [75]. Since the experiment was conducted during their maturation period, the dramatic hormonal fluctuations and immune responses during maturation may have had different effects on the astrocytic activity levels in the frontal cortex of the male and female rats. This maturation period may be the cause of the observed sex differences in pain response, as suggested in previous studies [76,77]. Nonetheless, GEG was able to alleviate the effects triggered by SNL-induced NP pain (Figure 5A and Figure 6A).

CD11b and IBA1 are commonly known as microglia markers that are located in the CNS [78]. In this study, SNL increased CD11b expression in males but reduced it in females. We found that GEG treatment downregulated CD11b in males in response to the elevated expression level caused by the SNL procedure. At the same time, an upregulation of CD11b levels in female rats by GEG treatment was observed in the SNL-treated female rats. Our previous study on male rats with diabetic NP [23] found that GEG administration suppressed CD11b gene expression in the amygdala and colon. In the present study, such inhibitory effects of GEG on CD11b gene expression in the amygdala were observed in the male (but not female) NP rats. This may be due to different mechanisms and signaling pathways involved in diabetes-induced NP and SNL-induced NP. The differential glial activation in response to NP pain in male and female rats also suggests sex differences in pain responses. Importantly, GEG showed beneficial effects regardless of sex in this study and was able to reduce the glial activity down to baseline levels.

In addition to the CNS, IBA1 is also expressed in other types of macrophages and cells involved in the immune response in non-neuronal tissues, such as the colon [79]. In this study, GEG decreased the expression of IBA1 in the amygdala of male and female NP rats, which is consistent with a previous study showing that 6-shogaol (a bioactive compound of ginger extract) inhibited microglial activation in the substantia nigra pars compacta in a Parkinson’s disease mouse model [80]. GEG administration to NP rats also significantly suppressed the IBA1 gene expression in the colon of both sexes, which agrees with our previous study that peanut shell extract with high antioxidant/anti-inflammatory properties decreased IBA1 in non-neuronal tissues, such as liver and white adipose tissues [41].

Mitochondrial biogenesis, along with fission and fusion, is vital for the maintenance of mitochondrial homeostasis, which plays a key role in neuronal development, synaptogenesis, plasticity, neural adaptations, and behavior [81]. PGC1α has been shown to regulate oxidative stress and maintain normal mitochondrial biogenesis and function in NP [82,83]. PGC1α promotes TFAM expression through nuclear respiratory factor 1/2 (NRF1/2). NRF1/2 and TFAM upregulate mitochondria-related proteins, including respiratory chain complexes, and induce the replication and transcription of mitochondrial DNA, respectively [82]. The PGC1α-NRF1/2-TFAM axis plays an important role in the regulation of mitochondrial regeneration. PGC1α, TFAM, and NRF1 were significantly downregulated in spinal neurons in an NP model due to upregulated levels of ROS and oxidative stress [84]. In the present study, GEG supplementation increased gene expressions of PGC1α and TFAM in the amygdala and frontal cortex of male and female NP rats, suggesting neuroprotective effects on mitochondrial biogenesis. Ginger extract, 6-gingerol, and 6-shogaol (ginger bioactive components) promoted mitochondrial biogenesis via the activation of the AMPK-PGC1α signaling pathway in non-neuronal tissues (namely, liver, muscle, and brown adipose tissues) of young male Balb mice [85]. In contrast to the amygdala and frontal cortex, we noted that GEG has a stimulatory effect on PGC1α gene expression but an inhibitory effect on TFAM in the colon, indicating perhaps different regulatory pathways of mitochondrial function in CNS versus peripheral or non-neuronal tissues.

DRP1 and FIS1 are involved in mitochondrial fission [86]. Fusion of adjacent mitochondria is regulated by the MFN1 and MFN2 proteins, and by the optic atrophy 1 protein [86]. In this study, the effect of GEG supplementation on mitochondrial homeostasis, in terms of the balance between fission and fusion, seems to be tissue-specific. For example, GEG suppressed fission (FIS1 and DRP1) but enhanced fusion (MFN2 and MFN1) in the amygdala and colon of NP rats. On the other hand, GEG increased the gene expression of both fission (FIS1 and DRP1) and fusion (MFN2 and MFN2), along with TFAM and NRF1 in the frontal cortex of NP rats, further demonstrating GEG’s favorable mitochondrial biogenesis via increased mitochondrial turnover. This is the first study to elaborate on the beneficial impacts of GEG on mitochondrial homeostasis (biogenesis, fission, fusion, and mitophagy) in the brain and colon in NP, in part through its anti-inflammatory and antioxidant properties, as shown by increased Complex I or III mRNA expression as well as decreased TNFα and NFκB mRNA expression.

Translating the findings from animal models in this study to human conditions can be challenging. Our new clinical study in individuals with sciatic pain is designed to validate the observed effects (gut–brain axis) and to better understand the role of microbiota-gut–brain interactions by utilizing GEG supplementation in a randomized double-blinded, placebo-controlled trial in individuals with sciatic pain, a form of NP.

There are limitations to this study. As a note of caution, surgical stress during the procedure can affect the small intestine and induce inflammation. In order to control for any effects of surgical stress on the small intestine and inflammation, all animals received either sham or SNL surgical procedures. The acute effects of surgical stress on the small intestine and inflammation were similar for all study animals and did not confound the assessment of GEG effects, since (i) all animals received either sham surgery or SNL surgery, (ii) this study was focused on the colon (large intestine) instead of the small intestine, and (iii) this was a long-term chronic GEG feeding study. In this study, we did not measure the presence of GEG compounds in the blood. The GEG used in this study was composed of 6-, 8-, and 10-gingerol and 6-, 8-, and 10-shogaol. 6-Gingeral, 8-gingerol, and 6-shogal have been reported to have excellent BBB permeability and cross into the brain via passive diffusion [17]. As a note of caution, the number of animals per group was rather small, though sufficient to detect differences, but future studies with large sample sizes should validate these findings. Our study was not designed to fully determine the molecular mechanisms by which GEG exerts its effects, which should be investigated in future mechanistic studies.

## 5. Conclusions

GEG supplementation alleviated the emotional/affective pain behaviors of males and females via improving gut integrity, suppressing neuroinflammation, and improving mitochondrial homeostasis in the amygdala, frontal cortex, hippocampus, and colon in both male and female SNL rat models of NP, implicating the gut–brain axis in NP. Sex differences observed in the vocalization assays may suggest different mechanisms of evoked NP responses in females.

## Figures and Tables

**Figure 1 nutrients-16-03563-f001:**
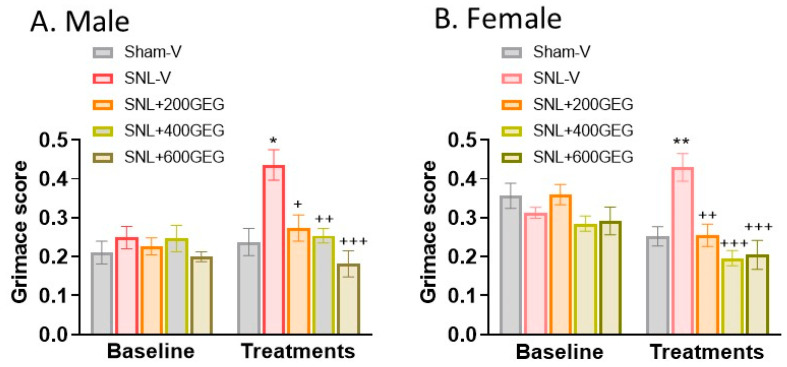
Effects of GEG on spontaneous pain (RGS score) in the male rats (**A**) and female rats (**B**). Data are expressed as mean ± SEM and were analyzed by one-way ANOVA followed by Bonferroni multiple comparisons test, *n* = 9 per group. * *p* < 0.05, ** *p* < 0.01 for SNL-V vs. Sham-V group. ^+^ *p* < 0.05, ^++^ *p* < 0.01, ^+++^ *p* < 0.001 SNL-GEG groups vs. SNL-V group.

**Figure 2 nutrients-16-03563-f002:**
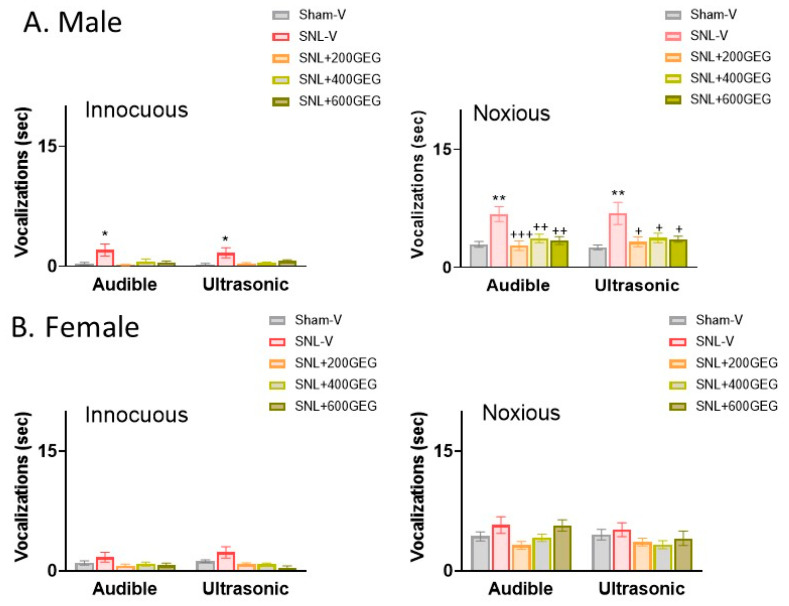
Effects of GEG on emotional pain responses (duration of vocalizations) in the male rats (**A**) and female rats (**B**). Data are expressed as mean ± SEM and were analyzed by one-way ANOVA followed by Bonferroni multiple comparisons test. *n* = 5–7 per group. * *p* < 0.05, ** *p* < 0.01 for SNL-V vs. Sham-V group. ^+^ *p* < 0.05, ^++^ *p* < 0.01, ^+++^ *p* < 0.001 for other groups vs. SNL-V group.

**Figure 3 nutrients-16-03563-f003:**
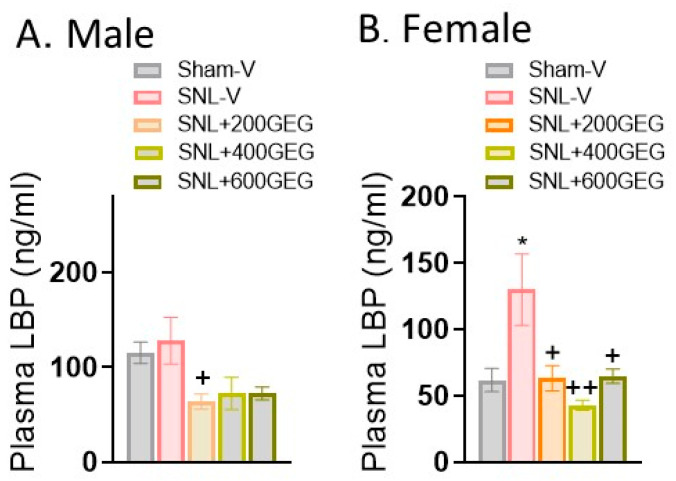
Effects of GEG on plasma LBP in the male rats (**A**) and female rats (**B**) assessed by ELISA. Data are expressed as mean ± SEM and were analyzed by one-way ANOVA followed by Bonferroni multiple comparisons test. *n* = 4–6 per group. * *p* < 0.05 compared with Sham-V group. ^+^ *p* < 0.05, ^++^ *p* < 0.01 compared with SNL-V group.

**Figure 4 nutrients-16-03563-f004:**
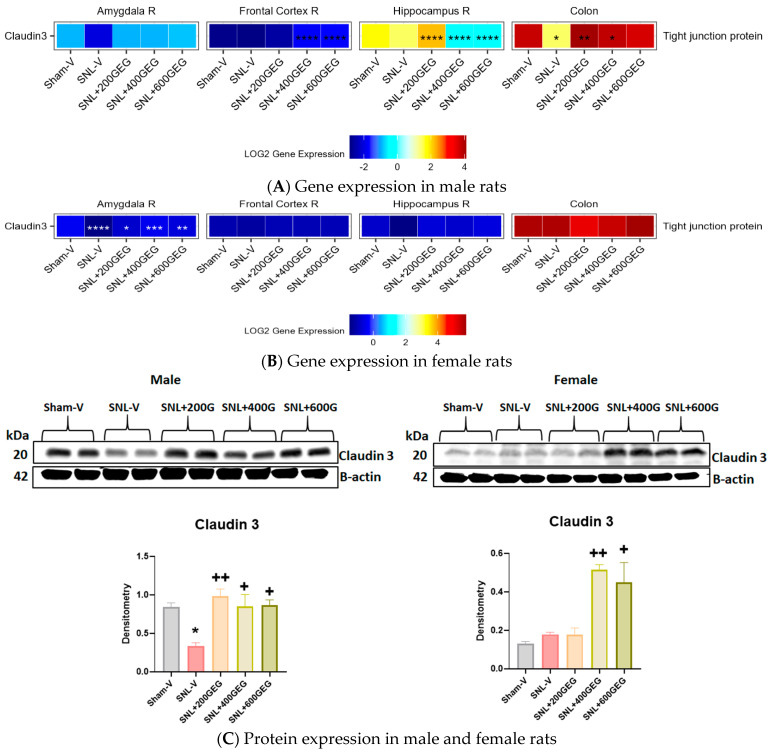
Effects of GEG on claudin 3 gene expression levels in amygdala, frontal cortex, hippocampus, and colon of male rats (**A**) and female rats (**B**) and protein expression levels in colon of male and female rats (**C**). For gene expression, data are expressed as mean ± SEM and were analyzed by one-way ANOVA followed by Tukey’s test, *n* = 7–9 per group. * *p* < 0.05, ** *p* < 0.01, *** *p* < 0.001, **** *p* < 0.0001 for SNL-V vs. Sham-V group, and other groups vs. SNL-V group. For protein expression, data are expressed as mean ± SEM and were analyzed by one-way ANOVA followed by Tukey’s multiple comparisons test, *n* = 7–9 per group. * *p* < 0.05 compared with Sham-V group. ^+^ *p* < 0.05, ^++^ *p* < 0.01 compared with SNL-V group.

**Figure 5 nutrients-16-03563-f005:**
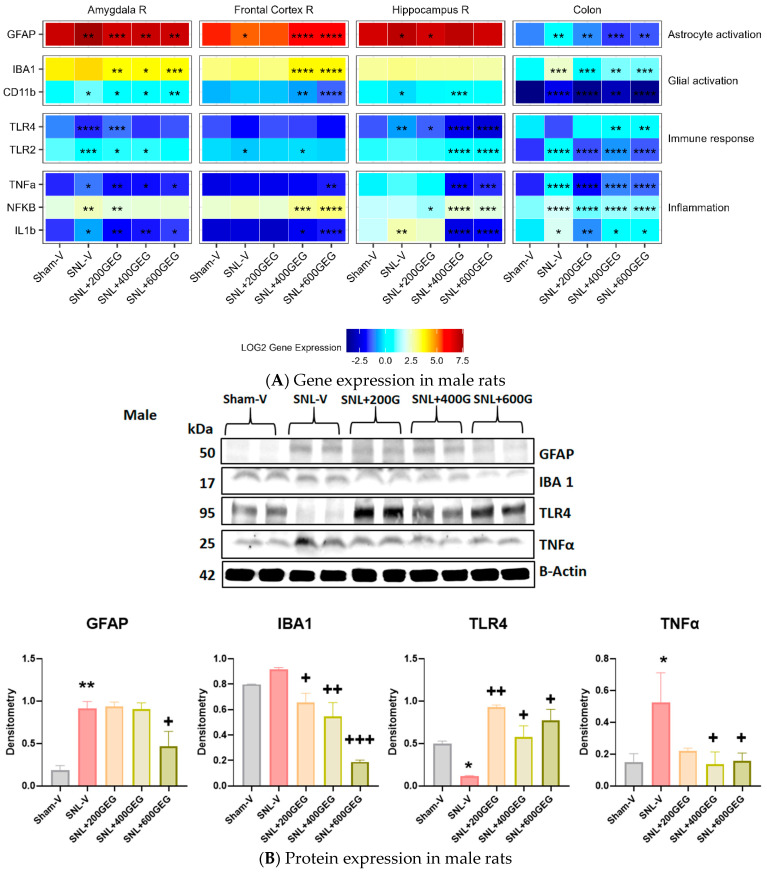
Effects of GEG on the neuroinflammation-associated gene expression levels in amygdala, frontal cortex, hippocampus, and colon of male rats (**A**) and protein expression levels in colon of male rats (**B**). For gene expression, data are expressed as mean ± SEM and were analyzed by one-way ANOVA followed by Tukey’s test, *n* = 7–9 per group. * *p* < 0.05, ** *p* < 0.01, *** *p* < 0.001, **** *p* < 0.0001 for SNL-V vs. Sham-V group, and other groups vs. SNL-V group. For protein expression, data are expressed as mean ± SEM and were analyzed by one-way ANOVA followed by Tukey’s multiple comparisons test, *n* = 7–9 per group. * *p* < 0.05, ** *p* < 0.01 compared with Sham-V group. ^+^ *p* < 0.05, ^++^ *p* < 0.01, ^+++^ *p* < 0.001 compared with SNL-V group.

**Figure 6 nutrients-16-03563-f006:**
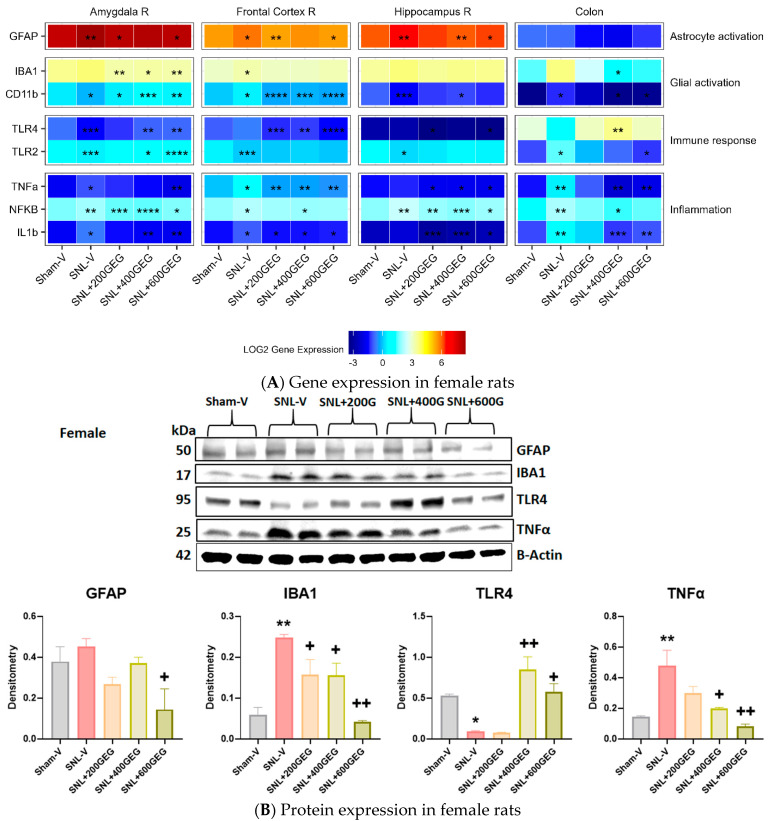
Effects of GEG on the neuroinflammation-associated gene expression levels in amygdala, frontal cortex, hippocampus, and colon of female rats (**A**) and protein expression levels in colon of female rats (**B**). For gene expression, data are expressed as mean ± SEM and were analyzed by one-way ANOVA followed by Tukey’s test, *n* = 7–9 per group. * *p* < 0.05, ** *p* < 0.01, *** *p* < 0.001, **** *p* < 0.0001 for SNL-V vs. Sham-V group, and other groups vs. SNL-V group. For protein expression, data are expressed as mean ± SEM and were analyzed by one-way ANOVA followed by Tukey’s multiple comparisons test, *n* = 7–9 per group. * *p* < 0.05, ** *p* < 0.01 compared with Sham-V group. ^+^ *p* < 0.05, ^++^ *p* < 0.01 compared with SNL-V group.

**Figure 7 nutrients-16-03563-f007:**
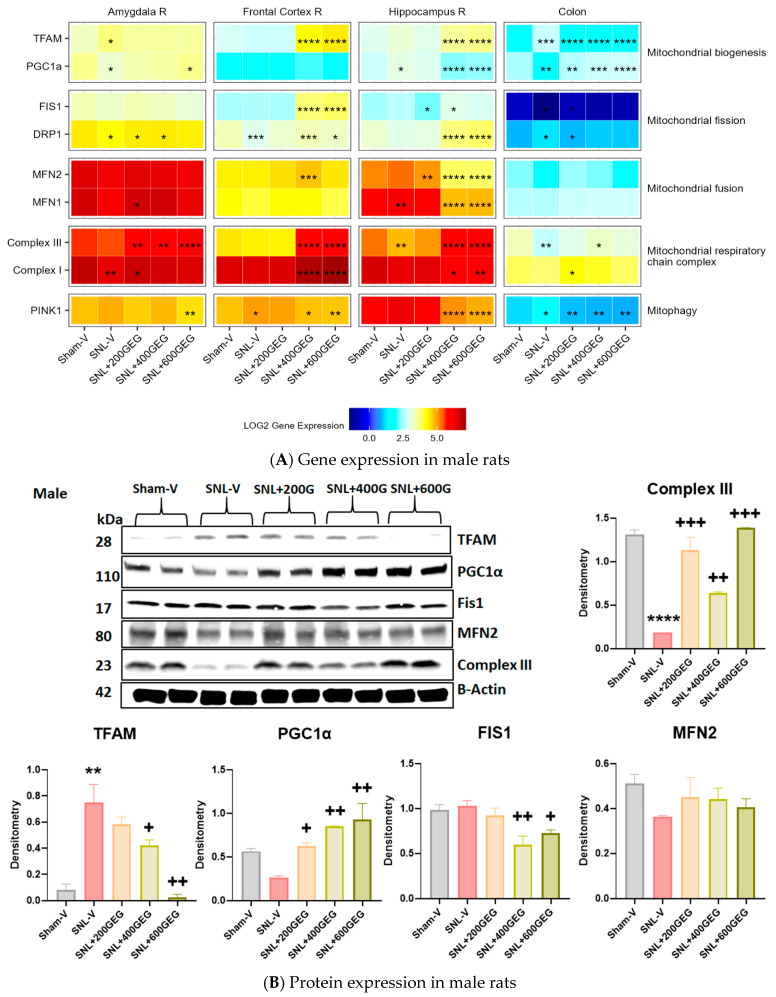
Effects of GEG on mitochondrial function-associated gene expression levels in amygdala, frontal cortex, hippocampus, and colon of male rats (**A**) and protein expression levels in colon of male rats (**B**). For gene expression, data are expressed as mean ± SEM and were analyzed by one-way ANOVA followed by Tukey’s test, *n* = 7–9 per group. * *p* < 0.05, ** *p* < 0.01, *** *p* < 0.001, **** *p* < 0.0001 for SNL-V vs. Sham-V group, and other groups vs. SNL-V group. For protein expression, data are expressed as mean ± SEM and were analyzed by one-way ANOVA followed by Tukey’s test, *n* = 7–9 per group. ** *p* < 0.01, **** *p* < 0.0001 compared with Sham-V group. ^+^ *p* < 0.05, ^++^ *p* < 0.01, ^+++^ *p* < 0.001 compared with SNL-V group.

**Figure 8 nutrients-16-03563-f008:**
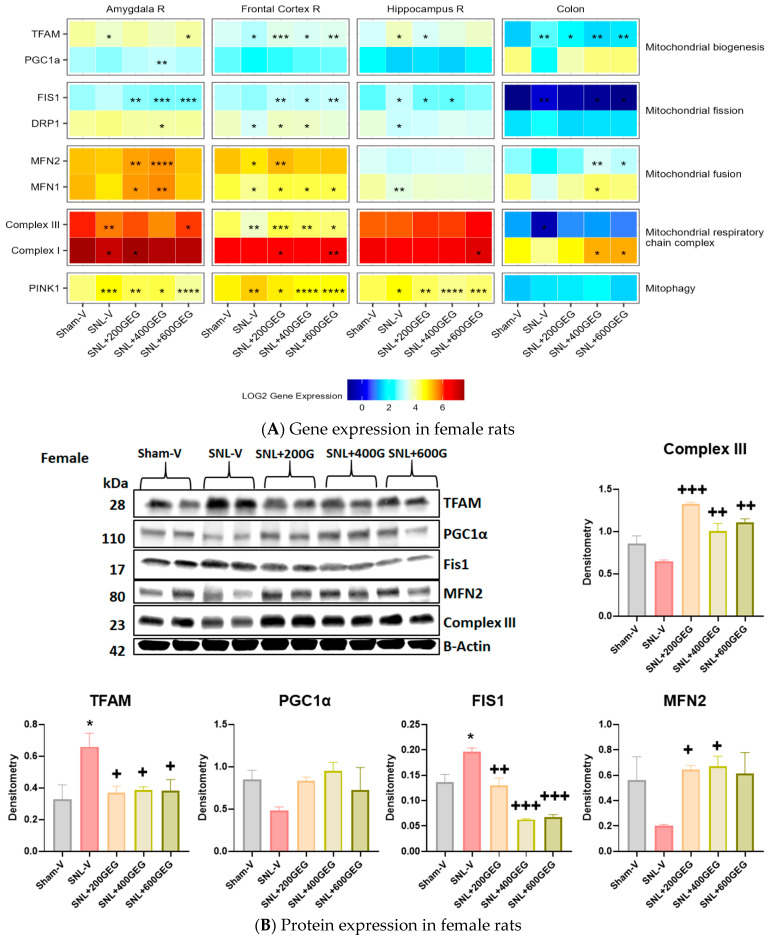
Effects of GEG on the mitochondrial function-associated gene expression levels in amygdala, frontal cortex, hippocampus, and colon of female rats (**A**) and protein expression levels in colon of female rats (**B**). For gene expression, data are expressed as mean ± SEM and were analyzed by one-way ANOVA followed by Tukey’s test, *n* = 7–9 per group. * *p* < 0.05, ** *p* < 0.01, *** *p* < 0.001, **** *p* < 0.0001 for SNL-V vs. Sham-V group, and other groups vs. SNL-V group. For protein expression, data are expressed as mean ± SEM and were analyzed by one-way ANOVA followed by Tukey’s multiple comparisons test, *n* = 7–9 per group. * *p* < 0.05 compared with Sham-V group. ^+^ *p* < 0.05, ^++^ *p* < 0.01, ^+++^ *p* < 0.001 compared with SNL-V group.

**Table 1 nutrients-16-03563-t001:** List of primers for mRNA expression.

Gene	Forward	Reverse
*Claudin 3*	5′-CCC AGC CTA CGG AGT TAC CC-3′	5′-TGC CGA TGA ATG CCG AAA CG-3′
*GFAP*	5′-AAT CTC ACA CAG GAC CTC GGC-3′	5′-AGC CAA GGT GGC TTC ATC CG-3′
*CD11b*	5′-TCC AAC CTG CTG AGG AAG CC-3′	5′-TCG ATC GTG TTG ATG CTA CCG-3′
*IBA1*	5′-GAG CTA TGA GCC AGA GCA AGG ATT T-3′	5′-ACT CCA TGT ACT TCG TCT TGA AGG-3′
*TLR4*	5′-TTG CAT CTG GCT GGG ACT CTG-3′	5′-TTC AGG GGG TTG AAG CTC AGA T-3′
*TLR2*	5′-AAG AGC ATC GGC TGG AGG TC-3′	5′-TGG AGC TGC CAT CAC ACA CA-3′
*TNFα*	5′-GAA CTC CAG GCG GTG TCT GT-3′	5′-CTG AGT GTG AGG GTC TGG GC-3′
*NFκB*	5′-CCT CCA CCC CGA CGT ATT GC-3′	5′-GCC AAG GCC TGG TTT GAG AT-3′
*IL1β*	5′-ATG TCT TGC CCG TGG AGC TT-3′	5′-ATG GGT CAG ACA GCA CGA GG-3′
*TFAM*	5′-GCT TCC AGG GGG CTA AGG ATG-3′	5′-TCG CCC AAC TTC AGC CAT TT-3′
*PGC1α*	5′-CAG GAG CTG GAT GGC TTG GG-3′	5′-GGG CAA AGA GGC TGG TCC T-3′
*FIS1*	5′-CTG CGG TGC AGG ATG AAA GAC-3′	5′-GGC GTA TTC AAA CTG CGT GCT-3′
*DRP1*	5′-ACA ACA GGA GAA GAA AAT GGA GTT G-3′	5′-AGA TGG ATT GGC TCA GGG CT-3′
*MFN2*	5′-TCC TGA ACA ACC GCT GGG AT-3′	5′-GAT CCA CCA CGC CTA GCT CA -3′
*MFN1*	5′-AGC TCG CTG TCA TTG GGG AG-3′	5′-TCC CTC CAC ACT CAG GAA GC-3′
*Complex III*	5′-GCA GTC CTC GCA TCC TAC CT-3′	5′-CTC CCG AGT GCT GTA GGC AT-3′
*Complex I*	5′-GGT TTG TCT ACA TCG GCT TCC-3′	5′-TAC AGA AGC TGG CGA TGC AAA-3′
*PINK1*	5′-TCG GCC TGT CAG GAG ATC CA-3′	5′-CAT TGC AGC CCT TGC CGA TG-3′
*β-actin*	5′-ACA ACC TTC TTG CAG CTC CTC C-3′	5′-TGA CCC ATA CCC ACC ATC ACA-3′

Abbreviations: *GFAP*, glial fibrillary acidic protein; *CD11b*, cluster of differentiation molecule 11b; *IBA1*, ionized calcium binding adaptor molecule 1; *TLR4*, toll-like receptor 4; *TLR2*, toll-like receptor 2; *TNFα*, tumor necrosis factor-α; *NFκB*, nuclear factor kappa-light-chain enhancer of activated B cells; *IL1β*, interleukin 1β; *TFAM*, mitochondrial transcription factor A; *PGC1α*, peroxisome proliferative activated receptor alpha; *FIS1*, fission 1; *DRP1*, dynamin-related protein 1; *MFN2*, mitofusin 2; *MFN1*, mitofusin 1; *PINK1*, PTEN-induced kinase 1.

**Table 2 nutrients-16-03563-t002:** Summary of antibody dilutions and conditions used in Western blot analysis.

Primary Antibody	Species	Dilution	Vendor	Secondary Antibody	Dilution	Vendor
Claudin 3	Rabbit polyclonal	1:1000	Invitrogen Thermo Fisher Scientific, Waltham, MA, USA	Goat anti-rabbit HRP	1:2000	Cell Signaling Technology, Inc., Danvers, MA, USA
GFAP	Mouse monoclonal	1:2000	Invitrogen Thermo Fisher Scientific, Waltham, MA, USA	Horse anti-mouse HRP	1:2000	Cell Signaling Technology, Inc., Danvers, MA, USA
IBA1	Rabbit monoclonal	1:1000	Cell Signaling Technology, Inc., Danvers, MA, USA	Goat anti-rabbit HRP	1:2000	Cell Signaling Technology, Inc., Danvers, MA, USA
TLR4	Mouse monoclonal	1:500	Novus Biological, Littleton, CO, USA	Goat anti-rabbit HRP	1:2000	Cell Signaling Technology, Inc., Danvers, MA, USA
TNFα	Rabbit polyclonal	1:1000	Abcam, Cambridge, MA, USA	Goat anti-rabbit HRP	1:2000	Cell Signaling Technology, Inc., Danvers, MA, USA
TFAM	Rabbit polyclonal	1:2000	Abcam, Cambridge, MA, USA	Goat anti-rabbit HRP	1:2000	Cell Signaling Technology, Inc., Danvers, MA, USA
PGC1α	Rabbit polyclonal	1:1000	Novus Biological, Littleton, CO, USA	Goat anti-rabbit HRP	1:2000	Cell Signaling Technology, Inc., Danvers, MA, USA
FIS1	Rabbit polyclonal	1:1000	Protein Tech Group, Inc., Chicago, IL, USA	Goat anti-rabbit HRP	FIS1	Cell Signaling Technology, Inc., Danvers, MA, USA
MFN2	Rabbit polyclonal	1:1000	Cell Signaling Technology, Inc., Danvers, MA, USA	Goat anti-rabbit HRP	1:2000	Cell Signaling Technology, Inc., Danvers, MA, USA
Complex III	Rabbit monoclonal	1:1000	Cell Signaling Technology, Inc., Danvers, MA, USA	Goat anti-rabbit HRP	1:2000	Cell Signaling Technology, Inc., Danvers, MA, USA
β-actin	Mouse monoclonal	1:2000	Millipore Sigma, Burlington, MA, USA	Horse anti-mouse HRP	1:2000	Cell Signaling Technology, Inc., Danvers, MA, USA

Abbreviations: GFAP, glial fibrillary acidic protein; IBA1, ionized calcium binding adaptor molecule 1; TLR4, toll-like receptor 4; TNFα, tumor necrosis factor-α; TFAM, mitochondrial transcription factor A; PGC1α, peroxisome proliferative activated receptor alpha; FIS1, fission 1; MFN2, mitofusin 2.

## Data Availability

The original contributions presented in the study are included in the article, further inquiries can be directed to the corresponding author.

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
