# Peer review of "Beneficial Effects of Ginger Root Extract on Pain Behaviors, Inflammation, and Mitochondrial Function in the Colon and Different Brain Regions of Male and Female Neuropathic Rats: A Gut–Brain Axis Study"

_nutrients, 2024, doi:10.3390/nu16203563_

Round 1
Reviewer 1 Report
Comments and Suggestions for Authors
The surgical stress during procedure could affect small intestine and induce inflammation? Do you analyze this situation?
When you said intestinal inflammation do you consider entire intestine or the small intestine, respectively the large intestine?
Do you check the presence of the compounds from ginger extract into the blood? Or if these compounds have the ability to pass from the intestine into the blood or from the blood into the brain?
The results are presented in the right form.
Author Response
Reviewer #1:
“The surgical stress during procedure could affect small intestine and induce inflammation? Do you analyze this situation?”
Response: Thank you for the comment. The surgical stress during procedure may affect small intestine and induce inflammation. While we did not analyze the surgical stress induced inflammation, we included a sham-operated group to control for any surgically induced changes, and we did find differences between these groups, suggesting that the neuropathic pain condition is fundamentally different from an acute post-surgical state. The goal of this study was to assess effects of ginger in colon and brain in neuropathic pain. We have included the following in Limitation of study of the Discussion.
“As a note of caution, surgical stress during the procedure could affect small intestine and induce inflammation. In order to control for any effects of surgical stress on small intestine and inflammation, all animals received either sham or SNL surgical procedure. The acute effects of surgical stress on small intestine and inflammation would be similar for all study animals and would not confound the assessment of GEG effects, since (i) all animals received either sham surgery or SNL surgery, (ii) this study was focused on colon (large intestine) instead of small intestine, and (iii) this was a long-term chronic GEG feeding study.”
“When you said intestinal inflammation do you consider entire intestine or the small intestine, respectively the large intestine?”
Response: Here we studied the colon, which forms the large intestine.
“Do you check the presence of the compounds from ginger extract into the blood? Or if these compounds have the ability to pass from the intestine into the blood or from the blood into the brain?”
Response: Thank you for the excellent comment. In this study, we did not check the presence of compounds from ginger extract in the blood. In the Method section, we stated that “Based on the results of gas chromatography-mass spectrometry, GEG in our study is made up of 18.7% 6-gingerol, 1.81% 8-gingerol, 2.86% 10-gingerol, 3.09% 6-shogoal, 0.39% 8-shogaol, and 0.41% 10-shogaol.” Simon et al. reported that 6-gingeral, 8-gingerol, and 6-shogal have excellent blood-brain barrier permeability to cross the blood-brain barrier via passive diffusion (Simon 2020). We have included the following under Limitation of study in the Discussion.
“In this study, we did not measure the presence of GEG compounds in the blood. GEG used in this study is composed of 6-, 8-, 10-gingerol and 6-, 8-, and 10-shogaol. 6-gingeral, 8-gingerol, and 6-shogal have been reported to have excellent blood-brain barrier permeability and cross into the brain via passive diffusion (Simon 2020).”
Ref: Simon A, Darcsi A, Kéry Á, Riethmüller E. Blood-brain barrier permeability study of ginger constituents. J Pharm Biomed Anal. 2020 Jan 5;177:112820. doi: 10.1016/j.jpba.2019.112820. Epub 2019 Aug 19. PMID: 31476432.
Reviewer 2 Report
Comments and Suggestions for Authors
The manuscript by Julianna Maria Santos is interesting and informative. I have the following questions and comments:
1, generally for the study of gut-brain axis, the gut microbiota should be involved. Have the authors checked the changes of the gut microbiota in response to ginger extract treatment? Why or why not?
2, for all the figures in the manuscript, I suggest the authors add an individual label (A, B, C, etc.) for all the panels. This would help the readers to understand the figures more easily.
3, the format of the figures are not consistent. Some bars have dots while others do not. The authors should revise.
4, the limitations of the study have not been discussed. The authors should revise.
5, the tables should be in a three-line format.
6, the number of the animals in figure 3 should be specified. Why only 4 rats were used in the SNL-V group?
Author Response
Reviewer #2:
“The manuscript by Julianna Maria Santos is interesting and informative.”
Response: Thank you for your kind words.
I have the following questions and comments:
1, generally for the study of gut-brain axis, the gut microbiota should be involved. Have the authors checked the changes of the gut microbiota in response to ginger extract treatment? Why or why not?
Response: Thank you for the comment. Yes, we have reported the changes of the gut microbiota in response to ginger extract treatment in the male NP rats (Shen et al. 2024). We included the following in the Introduction (page 3): “We recently reported that ginger polyphenols can reverse the molecular signature of amygdala neuroimmune signaling and modulate microbiome composition in neuropathic rats 22.”
Reference:
- Shen CL, Santos JM, Elmassry MM, Bhakta V, Driver Z, Ji G, Yakhnitsa V, Kiritoshi T, Lovett J, Hamood AN, Sang S, Neugebauer V. Ginger Polyphenols Reverse Molecular Signature of Amygdala Neuroimmune Signaling and Modulate Microbiome in Male Rats with Neuropathic Pain: Evidence for Microbiota-Gut-Brain Axis. Antioxidants (Basel). 2024 Apr 23;13(5):502. doi: 10.3390/antiox13050502. PMID: 38790607; PMCID: PMC11118883.
In this revision, we now added more information in the Introduction section:
“GEG-treated NP animals had an increased abundance of Flavonifactor, Hungatella, Anaerofustis stercorihominis, and Clostridium innocuum group, while they had a decreased abundance of Rikenella, Muribaculaceae, Clostridia UCG-014, Mucispirillum schaedleri, RF39, Acetatifactor, and Clostridia UCG-009 22.”
2, for all the figures in the manuscript, I suggest the authors add an individual label (A, B, C, etc.) for all the panels. This would help the readers to understand the figures more easily.
Response: Thank you for the comment. In this revision, we have added A, B, C, etc. in Figures 1, 2, and 3 and results and figure legends.
3, the format of the figures are not consistent. Some bars have dots while others do not. The authors should revise.
Response: Thank you for the comment. We revised the format of the figures for consistency.
4, the limitations of the study have not been discussed. The authors should revise.
Response: Thank you for the comment. We have included the limitations of the study in the Discussion.
5, the tables should be in a three-line format.
Response: Thank you for the comment. Tables 1 and 2 are now in a three-line format.
6, the number of the animals in figure 3 should be specified. Why only 4 rats were used in the SNL-V group?
Response: Thank you for the comment. The number of animals per group is now provided for the data in all figures. In Figure 3 (plasma LBP), the number of animals in the SNL-V group was limited due to the availability of plasma for the LPS test because some plasma had hemolysis issue and could not be used for analysis.
Reviewer 3 Report
Comments and Suggestions for Authors
"Beneficial effects of ginger extract on pain behaviours, inflammation, and mitochondrial function in the colon and different brain regions of male and female neuropathic rats: a gut-brain axis"
Summary:
The study investigates the effects of gingerol-enriched ginger (GEG) supplementation on neuropathic pain (NP) behaviours, neuroinflammation, and mitochondrial homeostasis in male and female rats. Using a spinal nerve ligation (SNL) model, researchers observed that GEG supplementation effectively reduced pain behaviours and inflammation and improved mitochondrial function across different brain regions and the colon, suggesting a potential role of the gut-brain axis in NP management.
Strengths:
1. Comprehensive Analysis: The study presents an in-depth investigation into the effects of GEG on NP by assessing multiple parameters, including pain behaviours, neuroinflammation, and mitochondrial homeostasis in several brain regions and the colon.
2. Sex-Specific Analysis: Including both male and female rats provides valuable insights into the sex-specific effects of GEG on NP, which is often overlooked in similar studies.
3. Use of Established Models: The study employs well-established methods such as the rat grimace scale, vocalization assessments, qRT-PCR, and western blot techniques, ensuring the reliability and validity of the results.
4. Gut-Brain Axis Focus: The investigation into the gut-brain axis provides novel insights into the potential mechanisms of GEG's impact on NP, emphasizing the importance of gut health in pain management.
Weaknesses:
1. Limited Human Relevance: While the study's findings are promising, translating results from animal models to human conditions can be challenging. Further studies in human subjects are necessary to validate the observed effects.
2. Sample Size: Although the study involved 100 rats, the number could still be considered moderate, particularly when dividing into different treatment groups. Larger sample sizes provide more robust data.
3. Lack of Detailed Mechanistic Insight: While the study demonstrates GEG's beneficial effects, it does not fully elucidate the underlying molecular mechanisms by which GEG exerts its effects, leaving room for further investigation.
Key Findings:
- GEG supplementation significantly reduced spontaneous pain behaviours in both male and female rats.
- Emotional-affective pain responses were decreased in male NP rats but not in females after GEG supplementation.
- GEG improved gut integrity by reducing intestinal permeability and enhancing tight junction protein expression.
- GEG mitigated neuroinflammation markers (e.g., GFAP, CD11b, IBA1, TNFα, NFκB, IL1β) in the brain and colon of NP rats.
- Mitochondrial function improved with GEG supplementation, as evidenced by increased markers for mitochondrial biogenesis and fission and decreased markers for fusion and mitophagy.
Minor Comments:
- The Introduction needs more information about ginger, particularly its botanical origin, the regions where it is cultivated, and whether it is already known in traditional medicine worldwide and in America.
- Consider including an image of ginger in the Introduction. This visual aid can enhance the reader's understanding and engagement with the topic. A separate "Conclusion" section is needed in the text.
- Over 50% of the bibliographic references are from the last five years.
- Are these last sentences of the Introduction a hypothesis or a fact? "We noted differences between the sexes related to pain-associated behaviours, gene expression of neuroinflammation and mitochondrial function, and intestinal integrity in a GEG-dose response manner." Please clarify or remove these sentences.
Author Response
Reviewer #3:
Summary:
The study investigates the effects of gingerol-enriched ginger (GEG) supplementation on neuropathic pain (NP) behaviours, neuroinflammation, and mitochondrial homeostasis in male and female rats. Using a spinal nerve ligation (SNL) model, researchers observed that GEG supplementation effectively reduced pain behaviours and inflammation and improved mitochondrial function across different brain regions and the colon, suggesting a potential role of the gut-brain axis in NP management.
Strengths:
- Comprehensive Analysis: The study presents an in-depth investigation into the effects of GEG on NP by assessing multiple parameters, including pain behaviors, neuroinflammation, and mitochondrial homeostasis in several brain regions and the colon.
- Sex-Specific Analysis: Including both male and female rats provide valuable insights into the sex-specific effects of GEG on NP, which is often overlooked in similar studies.
- Use of Established Models: The study employs well-established methods such as the rat grimace scale, vocalization assessments, qRT-PCR, and western blot techniques, ensuring the reliability and validity of the results.
- Gut-Brain Axis Focus: The investigation into the gut-brain axis provides novel insights into the potential mechanisms of GEG's impact on NP, emphasizing the importance of gut health in pain management.
Response: Thank you for the appreciation of our work.
Weaknesses:
- Limited Human Relevance: While the study's findings are promising, translating results from animal models to human conditions can be challenging. Further studies in human subjects are necessary to validate the observed effects.
Response: Thank you for the excellent suggestion. We could not agree more on the need for translational research. In fact, based on our animal work, we recently received funding from the USDA NIFA (grant number: 2024-67018-42457) to translate our animal findings into a clinical trial in individuals with sciatic pain to better understand the role of microbiota-gut-brain interactions by utilizing GEG supplementation in a randomized double-blinded, placebo-controlled trial in individuals with NP. In this revision, we have included the following in the Discussion:
“Translating the results from animal models in this study to human conditions can be challenging. Our new clinical study in individuals with sciatic pain is designed to validate the observed effects (gut-brain-axis) and to better understand the role of microbiota-gut-brain interactions by utilizing GEG supplementation in a randomized double-blinded, placebo-controlled trial in individuals with sciatic pain, a form of NP.”
- Sample Size: Although the study involved 100 rats, the number could still be considered moderate, particularly when dividing into different treatment groups. Larger sample sizes provide more robust data.
Response: Thank you for the comment. We agree that a larger sample size would provide more robust data.
In this revision, we have included the following in the limitations of Discussion:
“As a note of caution, the number of animals in each treatment group is rather small, though sufficient to detect differences, but future studies with large sample sizes should validate these findings.”
- Lack of Detailed Mechanistic Insight: While the study demonstrates GEG's beneficial effects, it does not fully elucidate the underlying molecular mechanisms by which GEG exerts its effects, leaving room for further investigation.
Response: Thanks for this important observation. In this revision, we have included the following in the limitations of Discussion:
“Our study was not designed to fully determine the molecular mechanisms by which GEG exerts its effects, which should be addressed in future mechanistic studies.”
Minor Comments:
- The Introduction needs more information about ginger, particularly its botanical origin, the regions where it is cultivated, and whether it is already known in traditional medicine worldwide and in America.
Response: Thank you for the suggestion.
In this revision, we have included the following in the Introduction:
“Ginger (Zingiber officinale Rosc.), also known as adrak (Hindi), Jengibre (Spanish), Jiang (Chinese), or Zanjabeel (Arabic) is a perennial herbaceous plant that is grown across the Indian subcontinent, including India, Japan, and Indonesia, as well as in countries like Brazil and Jamaica. Ginger is best known for its culinary and medicinal uses. It belongs to the Zingiberaceae family, which includes turmeric, another plant well-known for its medicinal applications. The rhizome of ginger is marketed in various forms, including raw ginger, dry ginger, ginger powder, ginger oil, and ginger oleoresin. These products find their use in various foods such as ginger candy, ginger beer, ginger squash, ginger flakes, pickles, sweet vinegar or just as raw ginger powder. Ginger also is well known in the traditional medicinal systems of India (Ayurveda) and China (TCM), as well as in the Middle East and Africa. India is the world’s largest exporter of ginger powder, where ginger is cultivated across the country, with northeastern belt of India representing a major ginger growing area.”
- Consider including an image of ginger in the Introduction. This visual aid can enhance the reader's understanding and engagement with the topic. A separate "Conclusion" section is needed in the text.
Response: Thank you for the comment. In this revision, we have made the following:
- Title: We clarified that we used “ginger root extract”, instead of “ginger extract”. More information about ginger has been provided in the Introduction section.
- We do not believe that an image of ginger is appropriate for this research article on the effects of ginger root extract. It may be better suited for a review article on ginger.
- A separate “5. Conclusion” section has now been added.
- Over 50% of the bibliographic references are from the last five years.
Response: Thank you for the comment. The reason is that we wanted to make sure to provide the most up-to-date information. If we missed any older key reference, please let us know.
- Are these last sentences of the Introduction a hypothesis or a fact? "We noted differences between the sexes related to pain-associated behaviours, gene expression of neuroinflammation and mitochondrial function, and intestinal integrity in a GEG-dose response manner." Please clarify or remove these sentences.
Response: Thank you for catching that. The statement refers to the finding of this study. Therefore, this sentence has been removed now.
Round 2
Reviewer 2 Report
Comments and Suggestions for Authors
The authors have revised the manuscript accordingly. It can be considered for publication.